# Attempt to Optimize the Corrosion Resistance of HRB400 Steel Rebar with Cr and RE

**DOI:** 10.3390/ma15228269

**Published:** 2022-11-21

**Authors:** Tao Liu, Nannan Li, Chao Liu, Jingshe Li, Tianyi Zhang, Xuequn Cheng, Shufeng Yang

**Affiliations:** 1School of Metallurgical and Ecological Engineering, University of Science and Technology Beijing, Beijing 100083, China; 2Corrosion and Protection Center, Institute for Advanced Materials and Technology, University of Science and Technology Beijing, Beijing 100083, China; 3National Materials Corrosion and Protection Data Center, University of Science and Technology Beijing, Beijing 100083, China; 4School of Energy Power and Mechanical Engineering, North China Electric Power University, Beijing 100096, China

**Keywords:** HRB400 steel rebar, Cr, RE, corrosion resistance

## Abstract

The corrosion resistance of the HRB400 steel rebar alloyed with Cr and rare earths (RE) was systematically studied from two aspects, including the properties of the passive film and the protectiveness of the rust layer. The results presented that Cr increased the corrosion resistance of the steel rebar through stabilizing the passive film and was not involved in the formation of corrosion pits, while the pitting corrosion was initiated by the dissolution of (RE)_2_O_2_S inclusion, resulting in the local acidification at the bottom of the corrosion pits, which decreased the stability of the passive film. As for the long-term corrosion process, both Cr and RE decreased the corrosion rate of the steel rebar, which was related to the promotion effect on the formation of α-FeOOH in the rust layer from Cr and RE.

## 1. Introduction

HRB400 steel rebars are widely used in reinforcing concrete structures such as buildings and bridges because of the relatively high strength for the same cost, consistent quality, and good mechanical and welding properties. However, during the service phase of reinforced concrete, as it is put into use over time, the inevitable porosity of the mesh structure in concrete leaves room for the diffusion of aggressive substances from outside [1,2,3,4,5]. These defects can react with the reinforced concrete and consume the base material, causing steel rebars to rust and further develop into structural failure, reducing the strength and aesthetics of the reinforced concrete. This might result in huge economic losses and dangerous safety issues that cannot be ignored [6,7,8,9,10].

Microalloying is a cheap and effective way of improving the corrosion resistance of low-alloy steels. It has been shown that the alloying of Cr and rare earths (RE) can improve the corrosion resistance of low-alloy steels. The research on the effect of Cr on the rust layer is the most sufficient. Cr can improve the uniform corrosion resistance of steel, because Cr elements can gather in the inner rust layer to make the inner rust layer denser, and change the proportion of Fe oxides in the inner rust layer [11,12,13]. At the same time, Cr is thought to improve the protective properties of passivated films by improving the oxide and hydroxide content of the film. RE are considered effective for both improving mechanical properties and reducing the size of inclusions. RE can soften high-hardness inclusions through the formation of (RE)_2_O_2_S and (RE)AlO_3_ and reduce inclusion-induced lattice distortion [14,15,16,17]. In addition, RE as deoxidants improve the pitting resistance of steels. Therefore, the mechanisms of pitting corrosion after modification with rare earths are complex and varied. Specifically, the matrix surrounding the Al_2_O_3_ inclusions in the steel is selectively dissolved and craters are formed, whereas after modification with RE, the modified inclusions are preferentially dissolved and craters are formed [18,19]. Despite the large number of studies describing the effect of Cr and RE on the corrosion resistance of low-alloy steels, the mechanism of the synergistic effect of Cr and RE on the corrosion resistance of low-alloy steels is still not adequately studied. There is also a paucity of reports on the application of microalloying methods for low-alloy steels in the production process of steel rebars.

With the increasing prominence of the above problems, more and more attention is being paid to the study of the corrosion behavior of steel rebars. Based on this, this experiment has selected the conventional steel rebar HRB400 as the object of study, the effect of Cr and RE on the microstructure was studied using optical microscopy, the morphology of inclusions, localized corrosion morphology, and surface and cross-sectional morphology of the rust layer were analyzed using scanning electron microscopy (SEM) and energy-dispersive spectrometer (EDS), the elemental valence of the passivation film was studied by using X-ray photoelectron spectroscopy (XPS), the phase of the rust layer was analyzed using X-ray diffraction (XRD), and the electrochemical properties of the passive film were characterized by using electrochemistry methods. The conclusion summarized the evolution of the corrosion resistance of HRB400 steel rebars after alloying Cr and RE, providing guidance on the manufacture of corrosion-resistant HRB400 steel rebars.

## 2. Experimental

### 2.1. Materials and Microstructure Observation

Three different types of hot-rolled HRB400 steel rebars made by Wuhu-Xinxing pipes Co. Ltd. (Wuhu, Anhui province, China), were investigated in this research. The chemical composition (wt.%) was 0.24% C, 0.6% Si, 1.52% Mn, 0.02% P, 0.03% S, and the residual Fe. Two kinds of steels were obtained by adding 0.5% Cr and 0.5% Cr + 0.012% RE (Ce + La) into the HRB400 steel rebar, respectively. The new alloy steels were respectively marked as Cr and RE. For microstructure observation, three steel rebars were cut into the block specimens with a size of 10 × 10 × 3 mm^3^. The microstructures were observed by the optical microscope after being polished with P5000 silicon carbide paper and treated with 4% nitric acid alcohol for 5 s [6,11]. The inclusions were directly characterized by SEM after polishing with P5000 silicon carbide paper.

### 2.2. Immersion Test

The three steel rebars were cut into block samples with a size of 10 × 10 × 5 mm^3^, and then immersed in the saturated Ca(OH)_2_ solution with 1 wt.% NaCl (simulated solution) for 10 min, 40 min, and 4 h, and then descaled with rust removal solution (500 mL HCl (37%) + 500 mL deionized water + 3.5 g C_6_H_12_N_4_), and finally observed by SEM and EDS [6,7,8]. XPS measurements were then carried out after immersing in the saturated Ca(OH)_2_ solution for 24 h for the composition analysis of the passive films on three steel rebars.

### 2.3. Electrochemical Measurements

A standard three-electrode system was used for electrochemical experiments to characterize the protectiveness of the passive film [15,16,17,18,19,20]. In the test, the specimen was the working electrode, the platinum sheet was the counter electrode, and the saturated glycerol electrode was the reference electrode. A sample with a size of 10 × 10 × 3 mm^3^ was epoxied to retain a working area of 1 cm^2^. Before the experiment, the samples were ground to P2000 with water sandpaper, and cleaned and blown dry with alcohol and deionized water. Electrochemical tests including cyclic potentiodynamic polarization and electrochemical impedance spectroscopy (EIS) were performed using a PARSTAT2273 electrochemical workstation (AMETEK, Inc., Berwyn, PA, USA). Potentiodynamic polarization tests were conducted from −1 to 1 V _vs. SCE_ at a scanning rate of 0.1667 mV/s. A stabilization time of 50 min was allowed prior to the EIS measurement. EIS measurements were performed over the frequency range from 100 kHz to 10 mHz with a 10 mV sinusoidal amplitude at open-circuit potential (OCP), and the data were fitted with ZSimpWin 3.3 software [20,21]. All tests were conducted in the Ca(OH)_2_ + 1 wt.% NaCl solution. All the experiments were repeated at least in triplicate within air at a relative humidity of approximately 25% and at room temperature.

### 2.4. Dry/Wet Cyclic Tests

The environment of the dry/wet cyclic test is the saturated Ca(OH)_2_ + 1 wt.% NaCl solution. Corrosion periods were set at 72, 168, and 260 h, with a dry/wet cycle consisting of a 42 min drying process and an 18 min soaking process. At the end of the test, the surface and cross-sectional morphologies of the rust layer and the elemental distribution were observed by SEM and EDS. The composition of the crystalline phase of the rusts was analyzed by the XRD test by using an UltimalVX spectrometer (SmartLab, Japan) with a range of 10–90° at a scanning rate of 4°/min. The corrosion rate was calculated after the rust-removing process within the rust-removing solution mentioned in Section 2.3. The formula to calculate the corrosion rate was as follows [7,8,13]:(1)$$V_{corr}=\frac{(W_{0}-W_{i})\times 8.76\times 10^{4}}{S\rho t}$$
where $v_{corr}$ is the corrosion rate (mm/y), $W_{0}$ is the initial weight of the steel (g), $W_{i}$ is the weight of the steel after removal of the rust (g), *S* is the sample exposed area (cm^2^), ρ is the density of the sample (g/cm^3^), and *t* is the corrosion period (h).

## 3. Results and Discussion

### 3.1. Microstructure Observation

Figure 1 exhibits the microstructures of the three steel rebars. It can be seen that all samples consisted of pearlite (black) and ferrite (white), and there was obvious band tissue in the three steel rebars. Besides, the pearlite content of the base material (BM) (Figure 1a) and the Cr (Figure 1b) presented little differences, and that of the RE (Figure 1c) was lower than the other two steel rebars. This can be attributed to the influence of RE on the organization refinement. Due to the excellent physicochemical properties of RE and their oxides, including unique electron structure, outstanding chemical activity, and large ionic radius, RE can cause metamorphism, strengthening, and purification to improve the properties of steel [5,6,7,8]. In addition, with the decrease of the pearlite in the RE, the band-like structure in the steel rebar also significantly decreased. The previous study indicated that the RE element could reduce the activity of carbon in ferrite and increase the ability of ferrite to dissolve carbon, thereby changing the relative content of ferrite and pearlite [20,21]. Eventually, it could improve the comprehensive properties of steel rebars such as strength and plasticity. The obvious potential difference between the pearlite and the ferrite would provide a sufficient corrosion driving force, accelerating the corrosion process [22,23].

### 3.2. Inclusion Observation

Figure 2 presents the morphology and element distribution of inclusions in these three steels. As for BM, it can be found that an inclusion with an irregular shape should be an MnS inclusion (Figure 2a), combined with the EDS results. Si and O can be found in the MnS inclusion, which should be related to the oxygen removal in the smelting process [14]. As for the Cr steel rebar, CaO, Al_2_O_3_, and SiO were found in the inclusion (Figure 2b), which caused a different shape from the inclusion in BM steel. As for the RE steel rebar, the inclusion presented a regular circle shape, including La and Ce rare-earth elements, which confirms the effects of the RE elements on the formation of an inclusion in steel (Figure 2c). Thus, the element composition should further influence the corrosion properties of steel.

### 3.3. Immersion Test Results

Figure 3 shows the surface morphologies and element distribution maps of the steel rebars after immersion in saturated Ca(OH)_2_ with 1 wt.% NaCl solution for 10 min. The corrosion pits initiated around the inclusions, which indicates the high-corrosion susceptibility of the area around the inclusions. Figure 3a reveals that the slender pit appeared on the surface of the BM. EDS results show the enrichment of Mn and S elements at the bottom of the corrosion pits. It can be speculated that the pits were formed on the MnS inclusions, which should be related to the chemical dissolution of MnS [14]. Due to the localized electrochemical property difference between MnS and the matrix, MnS with a lower Volta potential would dissolve preferentially [24,25]. As shown in Figure 3b, the size of the corrosion pits on the Cr rebar surface is similar to that in BM. Inclusions in the Cr rebar consist of MnO, MnS·Al_2_O_3_, CaO·MgO, and SiO_2_. Trace amounts of Ca, Mg, and Si elements were enriched in the corrosion pit. The corrosion pits on the surface of the RE bar were the smallest, with only the enrichment of Mn and S elements, which should be induced by MnS inclusions in the matrix. Thus, it can be deduced that RE can improve the corrosion properties of the MnS inclusion [26,27].

Further, Figure 4 exhibits the surface morphologies of the steel rebars after being immersed in the simulated solution for 40 min. The obvious corrosion pits with a large size can be found on the BM bar, which showed significant corrosion morphologies compared to the other two steel rebars. The elements in the corrosion pits are consistent with the 10 min immersion test results in the BM, which is related to the corrosion of MnS, CaO, and Al_2_O_3_. Meanwhile, there were some corrosion products at the bottom of the corrosion pits, which is consist with the results in Figure 3. The enrichment of elements on the BM and the Cr rebar presents similar results, which suggests the little effect of Cr on the inclusions of the HRB400. However, the size of the pits of the Cr steel rebar is much smaller than that of the BM bar, indicating that the addition of the Cr element significantly inhibited the formation of the corrosion pits. Compared with the BM and the Cr rebar, the corrosion pits of the RE rebar were very small, and the pits were mainly MnS, which is consist with the results in Figure 3. It can be found that the size of the corrosion pits was about 1.8 mm in the RE rebar, which is smaller than that of the BM and the Cr rebar, indicating that the addition of RE elements can effectively reduce the size of the inclusions in the steel rebar. Therefore, compared with Cr, the pit size induced by RE was further reduced, indicating that RE can effectively improve the pitting corrosion resistance of the steel rebar.

As shown in Figure 5, obvious ferrite and lamellar pearlite can be found on the steel surface after the 4 h immersion test, indicating the intergranular corrosion of the steel rebar in the simulated solution. The corrosion degree cannot be clearly distinguished among the three steel rebars in Figure 5. MnS inclusion in the BM bar was longer than that in the Cr and the RE steel rebars, leading to a corrosion pit with the length of 30 μm. The length of corrosion pits on the Cr and the RE steel rebars was about 16 μm, which is around half of the BM. Different from Figure 3 and Figure 4, some RE can be found at the bottom of the corrosion pits in Figure 5.

In conclusion, the size of the inclusion-induced pits on the BM bar was larger than that of the Cr and the RE rebars, which corresponds to the inclusion size observed in Figure 2. Besides, it also indicates that the size of inclusions decreased with the addition of Cr and RE into HRB400 steel rebars, thereby inhibiting the formation of large-sized pits to a certain extent and improving the pitting corrosion resistance of steel rebars. Hence, in an alkaline service environment, uniform corrosion is the main form of corrosion on the surface of the steel rebar. The local corrosion induced by inclusions develops slowly. Corrosive ions such as H^+^, HS^−^, and S^2−^ that formed resulted from the dissolution of MnS, and the RE-modified inclusion did not obviously promote the pitting process [28]. This might occur because these corrosive ions were rapidly neutralized by the high pH test solution, which prevented the acidification of the localized solution and the development of the localized corrosion.

### 3.4. Electrochemical Results and XPS Analysis of the Passive Film

The typical cyclic potentiodynamic polarization results of three steel rebars subjected to four kinds of solutions with different Cl^−^ contents are shown in Figure 6. It can be found that all cathodic polarization curves presented similar Tafel slopes, which were from −100 mV/decade to −120 mV/decade, indicating the water reduction dominating the cathodic reaction in the solutions with or without Cl^−^ [15,17,29]. Meanwhile, the anodic polarization curves of the three steel rebars showed different evolution trends in various environments. As for the blank environment (0 Cl^−^), all samples showed a stable passive area in the range of high overpotential, presenting different passive current densities. The left shift of the anodic polarization curve from BM to Cr suggests an enhanced resistance to corrosion by alloying Cr in the steel rebar [22]. The passive current density RE was higher than that of BM, illustrating a higher dissolution rate in the passive film with the addition of Cr and RE. With the increase of Cl^−^, an almost overlapped stable passive area was observed in the 0.1 Cl^−^, expect for a slightly increased passive current density of RE, illustrating little differences in passivation characteristics in this environment. Meanwhile, the stable passive area cannot be found in BM and RE, except for Cr in the 1 Cl^−^, which presented an activation process in the range of low overpotential. Furthermore, the activation process replaced the passivation process in the 3.5 Cl^−^ for all steel rebars, which confirmed the destruction effect on the passive film of Cl^−^ [20,30]. Based on the work in [31,32], the dissolution rate of passive film is directly related to the passive current density, and all current densities of steels rebars increase with the increase of Cl^−^ content, indicating the promoting effects on the dissolution of the passivation film of Cl^−^. BM alloyed with Cr presented a better corrosion resistance than BM, which should be related to the participation of Cr-containing compounds in the formation of the passive film [33]. However, the simultaneous addition of RE and Cr elements had opposite effects on the stability of the passive film for BM. It has been reported that RE can modify the properties of inclusions relying on its activity in the steel [34,35], from which it can be inferred that the high activity of RE decreased the stability of the passive film. The increased Cl^−^ content can significantly damage the stabilization of the passive film despite the addition of Cr in steel rebar, which can be confirmed by Figure 6d.

Further, the composition of the passive film after immersion in saturated Ca(OH)_2_ solution for 24 h was detected by the XPS test, as shown in Figure 7, which spontaneously reflects the stability and protectiveness of the passive film under the original state for the three steel rebars [36,37]. The forms of Fe, O, and Cr elements in the passive film were characterized, and the corresponding spectra were fitted based on the binding energy. The spectrum of Fe2p3/2 contains three constituent peaks, which can be divided into the FeOOH peak (711.9 ± 0.5 eV), the Fe_3_O_4_ peak (711.2 ± 0.2 eV), and the Fe_2_O_3_ peak (710.8 ± 0.1 eV). The spectrum of O1s is also split into three peaks, including O^2−^ (529.8 ± 0.1 eV), OH^−^ (531.2 ± 0.1 eV), and H_2_O (529.8 ± 0.1 eV) [22]. The spectrum of Cr 2p3/2 is composed of Cr_2_O_3_ (574.8 ± 0.1 eV) and Cr(OH)_3_ (577.4 ± 0.1 eV). In general, the higher the valence state of the metal element, the more difficult it is to continue to oxidize it [20,21]. The results show that Cr and Fe exist in the form of Cr^3+^, Fe^2+^, and Fe^3+^ in the passive film, respectively, which indicates the critical roles of Cr^3+^ and Fe^3+^ on the protectiveness of the passive film. The area integration of the peak represents the content of the compounds in the passive film, and the results can be seen in Figure 7d. As for Cr and RE steel rebars, the ratios of Cr_2_O_3_ and Cr(OH)_3_ are about 1, which suggests the weak effect of RE on the formation of Cr_2_O_3_ or Cr(OH)_3_) [22,38]. Meanwhile, the addition of RE decreased the ratio of Fe_2_O_3_/Fe_3_O_4_, which should be related to the high activity of RE on the formation of inclusions. Combined with Figure 5, though the properties of inclusions were modified by RE for the steel rebar, RE can be involved in the corrosion process and the formed corrosion products deposited at the bottom of the corrosion pits, confirming the high activity of RE [16]. Fe_3_O_4_ is considered as the mixture of FeO and Fe_2_O_3_, which contains Fe^2+^ and Fe^3+^, presenting a lower corrosion stability than Fe_2_O_3_ simply containing Fe^3+^ [39]. Thus, RE decreased the stability of the passive film for the Cr steel rebar.

Figure 8 presents the EIS results of the three steel rebars in solutions containing different amounts of Cl^−^, which were fitted by the equivalent electrical circuit inserted in Figure 8(a1). All Nyquist curves of samples are composed of a large capacitance arc in all test environments, which are consistent with the two overlapping time constants shown in corresponding Bode plots. The equivalent electrical circuit used was determined based on the properties of the passive film in the high-frequency region and the charge transfer reaction process at the interface between the solution and the passive film in the low-frequency region. Thus, the whole electrode/film/electrolyte system of the steel rebars in the passive range can be divided into three impedances, including the electrolyte/passive film interface impedance, the passive film impedance, and the passive film/matrix impedance, and the fitting circuit can be expressed as: *R*_s_(*Q*_f_(*R*_*f*_(*Q*_dl_*R*_dl_))). As for the parameters in the circuit, *R*s is the solution resistance, *Q*_f_ is the constant phase element (CPE) of the passive film for signifying the non-ideal capacitance, *R*_*f*_ is the resistance of the passive film, *Q*_dl_ is the CPE of the double-electrode layer, and *R*_dl_ is the charge transfer resistance. The fitted results of EIS data can be seen in Table 1. Further, the sum value of *R*_*f*_ and *R*_dl_ was defined as *R*_total_, and the results can be seen in Figure 9. It can be found that the *R*_total_ of the Cr steel rebar was the highest among the three steels, suggesting the increase of the passive film corrosion protectiveness and the increase of the charge transfer resistance by the addition of Cr in BM. Meanwhile, adding Cr and RE at the same time, the impedance of the steel rebar presented a similar level as the BM in different solutions containing Cl^−^. To be more specific, *R*_dl_ dominates the decrease of *R*_total_, indicating the promotion of RE on the charge transfer process. The thickness can reflect the protectiveness of the passive film, which can be obtained from the following formulas based on previous studies [29,40]:(2)d=Aεε0ceff
(3)ceff=Y0Rfn−1n
where d is the passive film thickness, A is the area of the passive film, ε and $\varepsilon_{0}$ are the dielectric constant (15.6) and the vacuum permittivity (8.854 × 10^−14^ F/cm), respectively, $c_{eff}$ is the effective capacitance, and $Y_{0}$ and n are the magnitude and the dispersion coefficient of *Q*_f_, respectively. It can be derived from the formulas mentioned above that *R*_*f*_ is positively correlated with d, which suggests that the passive film protectiveness can be enhanced with the increase of the thickness. The film formation ratio is also positively correlated with d and *R*_*f*_, and Figure 9 presents similar *R*_*f*_ values of BM and RE steel rebars, which indicate the little impacts of RE on the formation of the passive film. Combined with the XPS results (Figure 7), it can be concluded from the EIS tests (Figure 9) that the addition of RE promoted the formation of Fe^2+^ in the passive film and promoted the charge transport process at the double-electrode layer, which decreased the protectiveness of the passive film of the Cr steel rebar.

### 3.5. Corrosion Protectiveness of the Rust Layer after the Passive Breakdown Process

For characterizing the local micromorphology, the rust after dry/wet cyclic tests was observed by SEM, as shown in Figure 10. In the early stages of corrosion, i.e., after 3 days of dry/wet cyclic tests, a loose rust layer consisting of a granular structure and a lamellar structure was formed on the surface of BM and Cr steel rebars. The rust layer on the surface of the RE steel rebars contained only granular structures, and elemental analysis revealed that the granular structure had a low Cl content, while the lamellar structure had a high Cl content, as shown in Figure 10a,d,g. As the corrosion time increased, the rust layer of the three steel rebars transformed into a lumpy structure. After 7 days of dry/wet cyclic tests, a dense rust layer was formed on the surface of the three steel rebars. Elemental analysis revealed that the RE rebar had the lowest Cl content in the rust layer, followed by Cr steel rebars, while BM steel rebars had the highest Cl content in the rust layer. After 15 days of dry/wet cyclic tests, similar to the 7-day results, the BM steel rebar still had the highest Cl content in the rust layer, the Cr steel rebar the next, and the RE rebar the lowest. At the same time, it can be found that the rust layer of the BM steel rebar had a large number of cracks in the block structure, the rust layer of the Cr steel rebar also had a few continuous cracks on the surface, while the rust layer of the RE rebar was dense and had the least cracks.

The corrosion rate results can be seen in Figure 11. As the corrosion time increased, the corrosion rate decreased for the three steel rebars, which was related to the increased protectiveness of the rust layer. At the initial stage of corrosion, i.e., at 3 days of dry/wet cyclic tests, the three steel rebars exhibited high corrosion rates, which was because no rust layer with a dense structure had been formed on the surface of the three steel rebars at this time. At this point, the corrosion rate of the RE rebar was the lowest, followed by Cr, which is consistent with the above results. After 7 days of dry/wet cyclic tests, the rust layer with a dense structure was formed, and the corrosion rate of the three steel rebars showed a significant reduction. At this point, the RE rebar still had the lowest corrosion rate and Cr the second lowest, which combined with the analysis of Cl content on the surface of the rust layer in the previous section, suggests that the rust layer of the RE rebar had the best protection when a stable rust layer first started to form. After 15 days of dry/wet cyclic tests, as the progress of corrosion progressed, the corrosion rate of the three steel rebars continued to decrease, indicating that the structure of the rust layer became denser. At this time, RE still had the lowest corrosion rate, followed by Cr, suggesting that Cr and RE act on the rust layer to further improve the protective properties of the already dense structure.

For discussing the functions of Cr and RE on the rust layer’s protectiveness, the cross-sectional morphologies and element distribution mappings were characterized and presented in Figure 12. As the rust layer after 3 days was heterogeneous and thin, the rust layers after 7 and 15 days were chosen and further analyzed. For the rust layer after 7 days, the rust layers of the Cr and the RE rebars formed a layered structure containing the outer rust layer and the inner rust layer. In contrast, the rust layer of BM still did not form a dense inner rust layer. Thus, in the elemental analysis, it can be found that the rust layer of BM was entered and enriched with a large amount of Ca from the environment, whereas in the rust layers of Cr and RE, Ca was mainly concentrated in the outer rust layer and was present a low content. A large amount of Cl could be found in the rust layer of BM and an enriched zone of Cl was present at the interface between the rust layer and the substrate. No enriched zone was found in the rust layer of Cr, and the Cr enrichment zone was observed in the inner rust layer, but significant amounts of Cl were still present. In the case of RE, similar to Cr, the enriched zone of Cr in the inner rust layer was also observed, while La and Ce were diffusely distributed throughout the rust layer. Cl was enriched in the outer rust layer, which indicates that the rust layer had good protective properties [22,30].

As the dry/wet cyclic tests progressed, after 15 days, the 3 steel rebars formed a dense rust structure composed of the outer and inner rust layers, and as a result, the amount of Ca entering the rust layer from the environment decreased. With the appearance of the dense rust layer, the Cl content of the three steel rebars also decreased. However, there were still continuous enriched zones of Cl in the inner rust layer of BM. In the rust layer of the Cr rebar, Cr was diffusely distributed throughout the rust layer, with only a small amount of Cl enriched in the outer rust layer. In the rust layer of the RE rebar, Cr, La, and Ce were all diffusely distributed in the rust layer, with minimal Cl content, suggesting that the protection of the rust layer increased as the corrosion time increased, which should be related to the addition of RE in the Cr rebar [35,41].

To obtain the phase composition of the three steel rebars after the dry/wet cyclic tests, the XRD results of 7 and 15 days are presented in Figure 13. In both corrosion cycles, the crystalline phase of the rust layer consisted of α-FeOOH, γ-FeOOH, and Fe_3_O_4_ [20,21]. There was not a significant difference in the physical phase, which should be related to the low addition of alloying elements, while the intensity of the α-FeOOH increased with the addition of Cr and RE, which indicates that Cr and RE promoted the formation of α-FeOOH. Comparing the intensity and the integral area of the physical phase, it can be roughly seen that the highest levels of Fe_3_O_4_ were found over the whole corrosion process, which should be attributed to the inadequate oxidation reaction during the dry/wet corrosion test. By comparing the results at 7 and 15 days, it can be found that the content of α-FeOOH increased with the corrosion cycles, which was in contrast to the content of γ-FeOOH, indicating that the compactness of the rust layer densified with the corrosion cycles. Compared with the 7- and 15-day XRD results, it can be found that the addition of RE in the Cr rebar decreased the intensity of the Fe_3_O_4_ peaks (2*θ* = 71–75°), suggesting the inhibition on the formation of Fe^2+^. Combined with the results in Figure 12, it can be concluded that RE can block the invasion of Cl^−^ and promote the formation of the stabilized substance in the rust layer, which increases the protectiveness of the rust layer.

## 4. Conclusions

The effect of Cr and RE on the protectiveness of the passive film and the rust layer on HRB400 steel rebar (BM) has been investigated in this work. The main conclusions are as follows.

The addition of Cr and RE had little effect on the microstructure of the BM, while it modified the properties of the inclusions instead of MnS, resulting in corrosion pits with a small size after the immersion tests. The improvement from Cr on the corrosion resistance of steel rebar was the formation of a stabilized passive film. On the contrary, RE was involved in the process of pitting corrosion and reduced the stability of the passive film for the Cr steel rebar, which was related to the activation effect of RE on the charge transfer process.

Both Cr and RE can increase the protectiveness of the rust layer through promoting the formation of a dense rust layer and increasing the content of the stabilized substance in the rust layer. The addition of RE in the Cr rebar can effectively enhance the protectiveness of the rust layer by blocking the invasion of Cl^−^.

## Figures and Tables

**Figure 1 materials-15-08269-f001:**
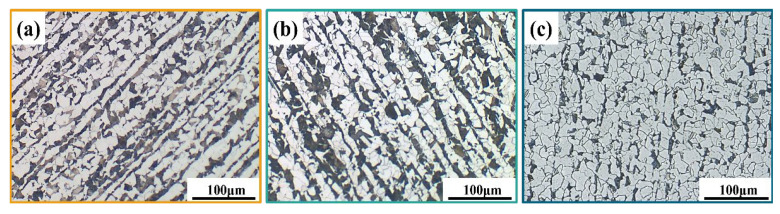
Microstructures of the tested steel rebars: (**a**) BM, (**b**) Cr, and (**c**) RE.

**Figure 2 materials-15-08269-f002:**
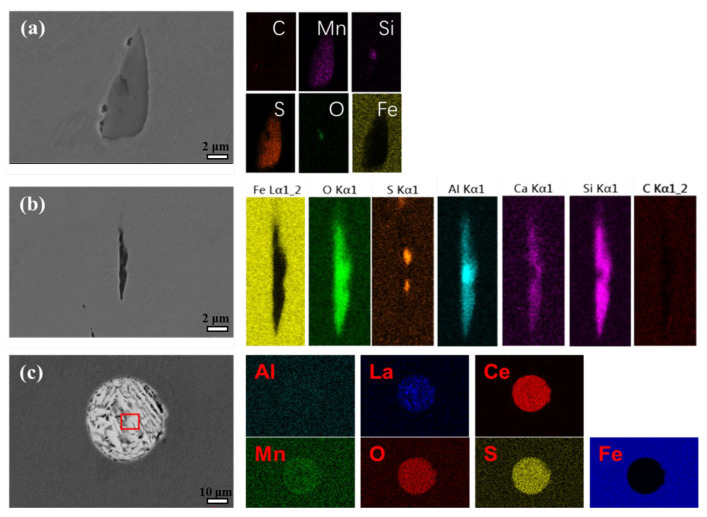
Observation and element distribution of inclusions in (**a**) BM, (**b**) Cr, and (**c**) RE steel rebars.

**Figure 3 materials-15-08269-f003:**
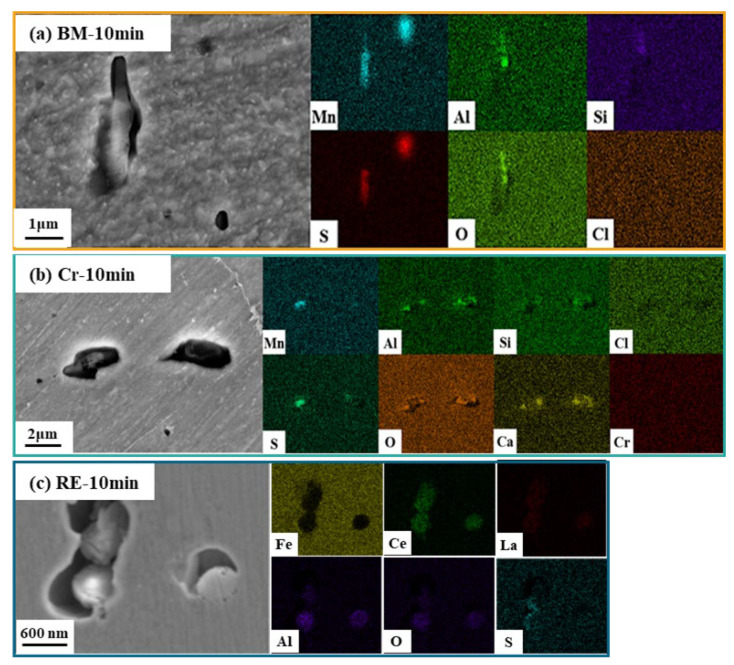
Immersion test results of (**a**) BM, (**b**) Cr, and (**c**) RE steel rebars after 10 min.

**Figure 4 materials-15-08269-f004:**
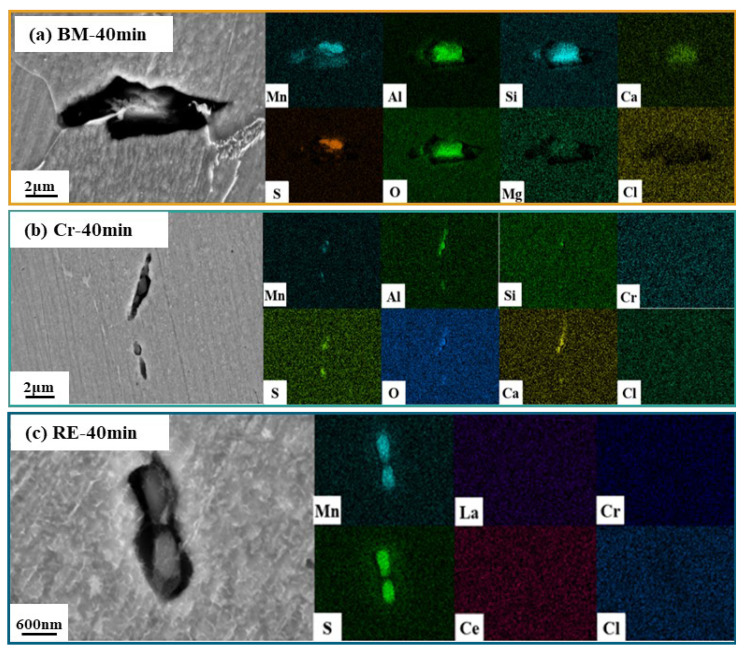
Immersion test results of (**a**) BM, (**b**) Cr, and (**c**) RE steel rebars after 40 min.

**Figure 5 materials-15-08269-f005:**
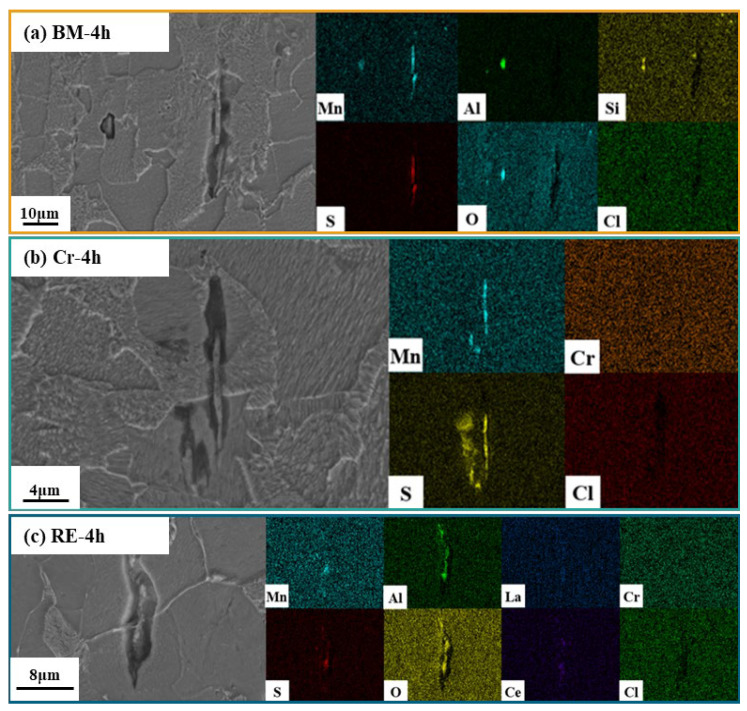
Immersion test results of (**a**) BM, (**b**) Cr, and (**c**) RE steel rebars after 4 h.

**Figure 6 materials-15-08269-f006:**
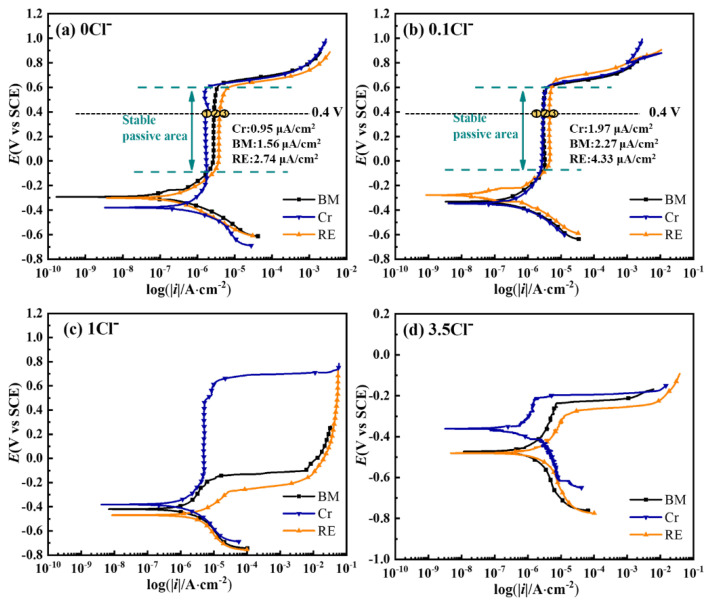
Cyclic potentiodynamic polarization curves of three steel rebars applied in different Cl^−^-containing solutions: (**a**) blank samples, (**b**) 0.1Cl^−^, (**c**) 1Cl^−^, and (**d**)3.5 Cl^−^ (unit: wt.%).

**Figure 7 materials-15-08269-f007:**
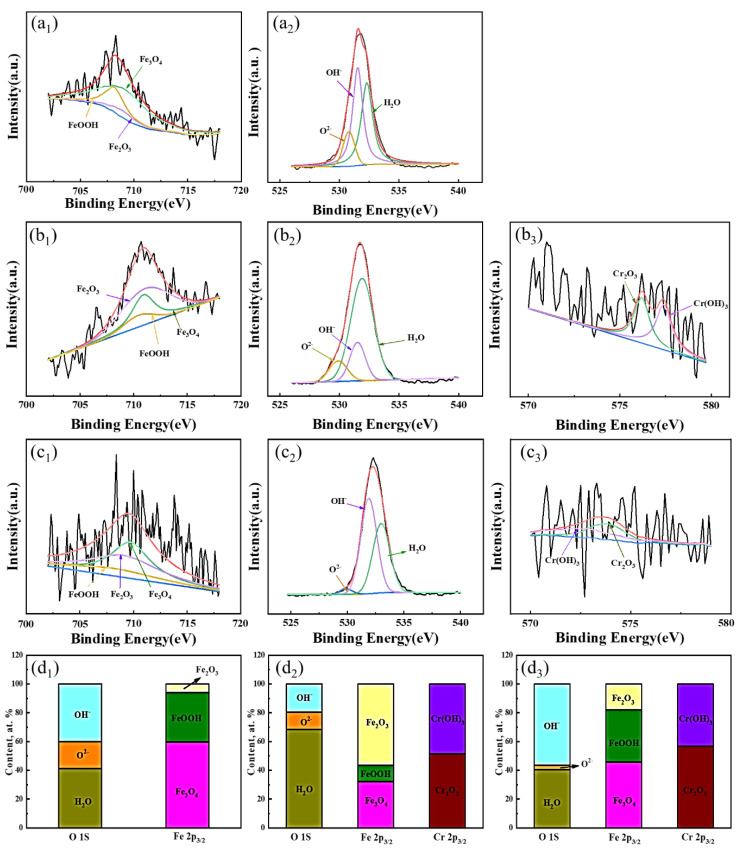
XPS results of Fe 2p3/2, Cr 2p3/2, and O 1 s, marked by 1, 2, and 3, of the passive film on (**a**) BM, (**b**) Cr, and (**c**) RE steel rebars formed in the saturated Ca(OH)_2_ solution for 24 h. (**d**) Content percentage of each substance.

**Figure 8 materials-15-08269-f008:**
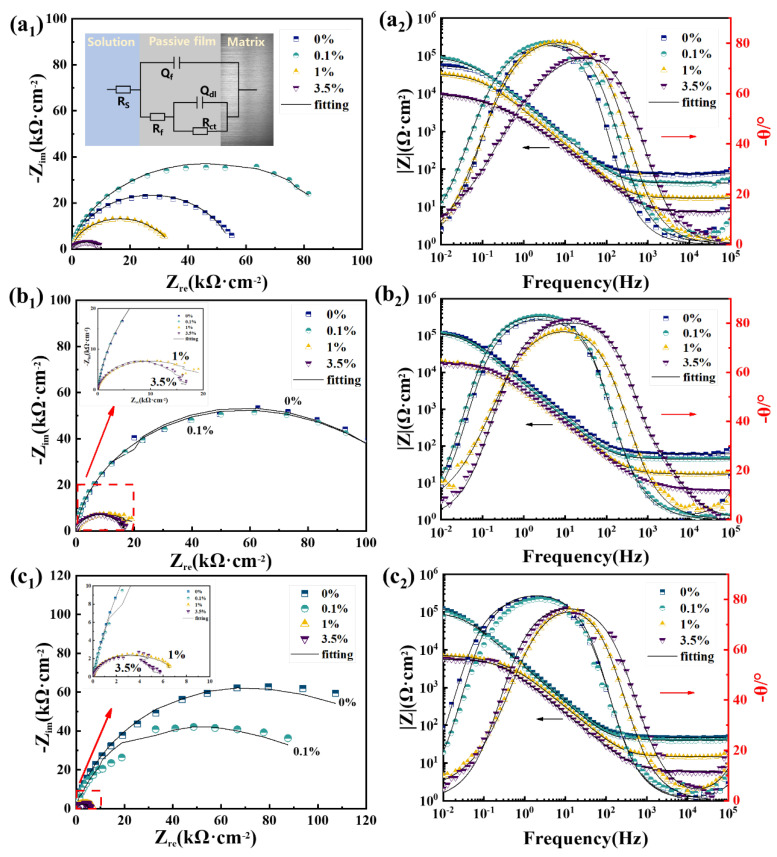
EIS plots of the passive film for (**a**) BM, (**b**) Cr, and (**c**) RE steel rebars in simulated solutions (1-Nyquist plots and 2-Bode plots). The equivalent electrical circuit for EIS data is inserted in (**a_1_**).

**Figure 9 materials-15-08269-f009:**
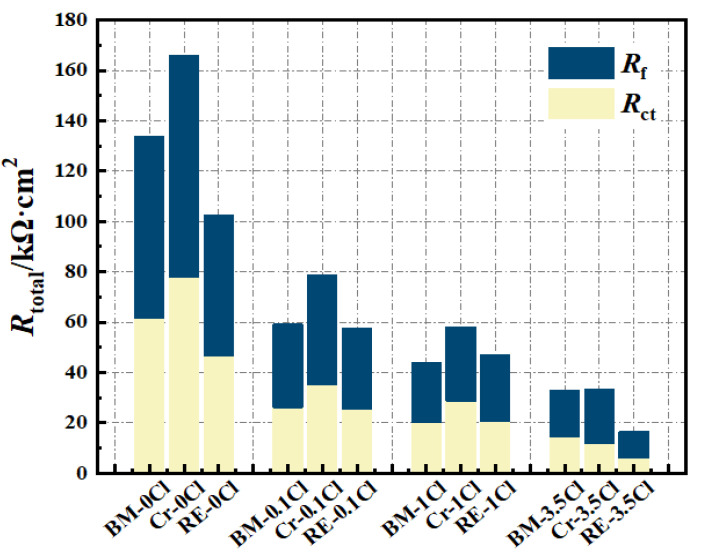
*R*_total_ (*R*_*f*_ + *R*_ct_) of the passive film for the three steel rebars in simulated solutions.

**Figure 10 materials-15-08269-f010:**
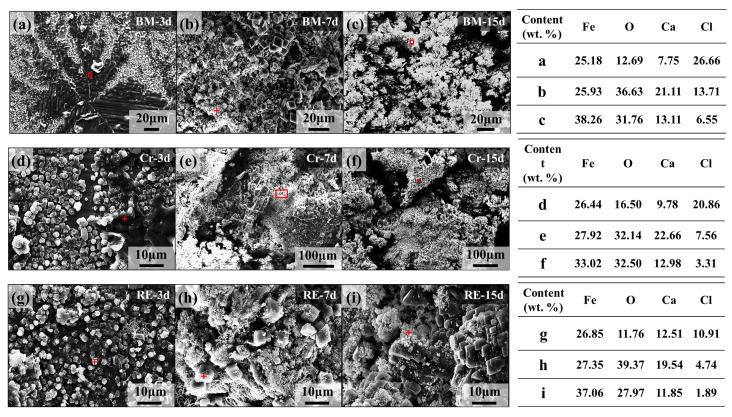
Surface morphologies and element distribution point scanning of the corrosion product film (CPF) of the three steel rebars after the corrosion process in the marine-reinforced concrete pore fluid simulation environment: (**a**) BM-3d, (**b**) BM-7d, (**c**) BM-15d, (**d**) Cr-3d, (**e**) Cr-7d, (**f**) Cr-15d, (**g**) Re-3d, (**h**) Re-7d, and (**i**) RE-15d. (“+” and “square” represent the sites of EDS test).

**Figure 11 materials-15-08269-f011:**
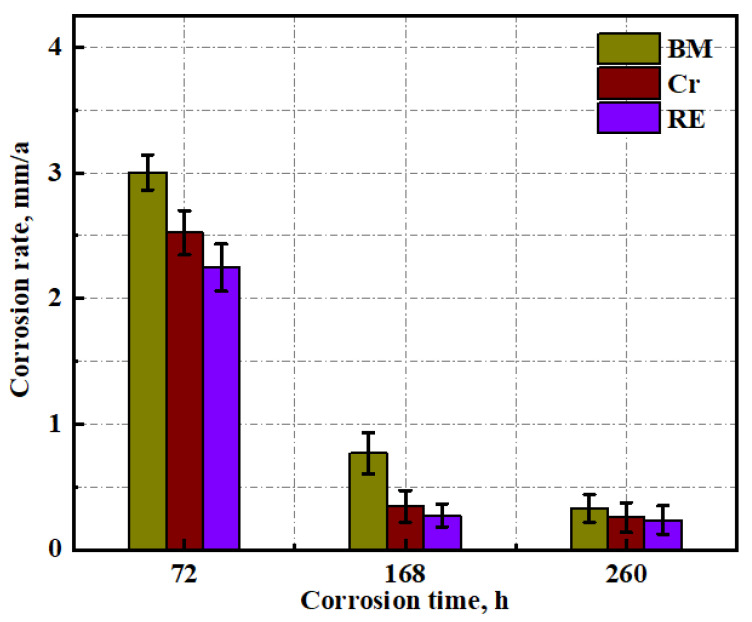
Corrosion rate of the three steel rebars after dry/wet cyclic tests.

**Figure 12 materials-15-08269-f012:**
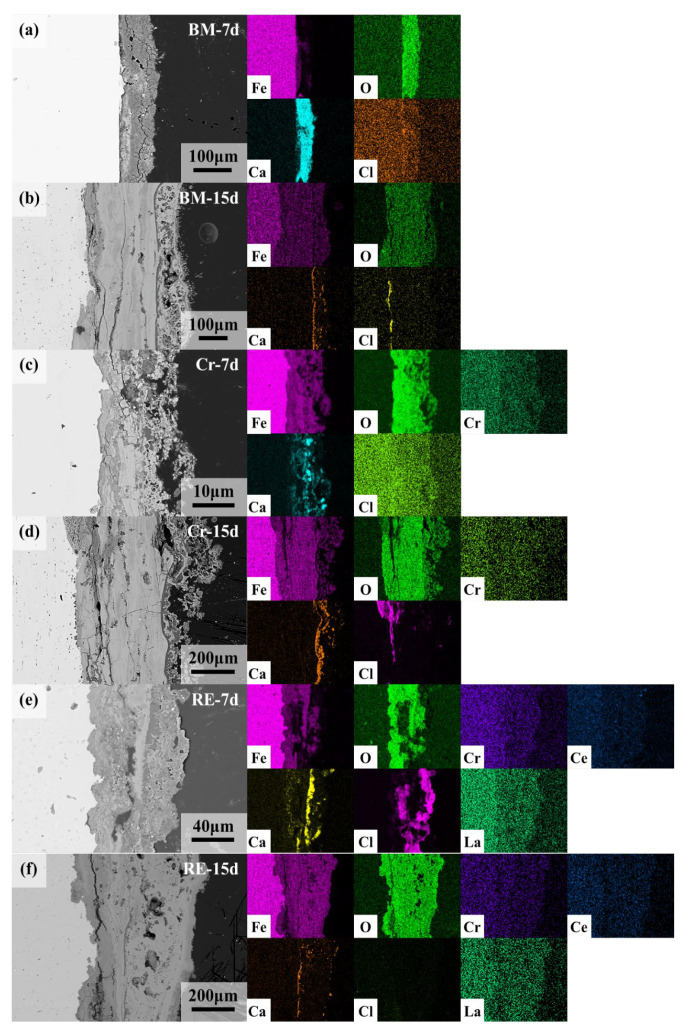
Cross-section morphologies and element distribution mappings of the corrosion product film (CPF) of the three steel rebars after the corrosion process in the marine-reinforced concrete pore fluid simulation environment: (**a**) BM-7d, (**b**) BM-15d, (**c**) Cr-7d, (**d**) Cr-15d, (**e**) Re-7d, and (**f**) RE-15d.

**Figure 13 materials-15-08269-f013:**
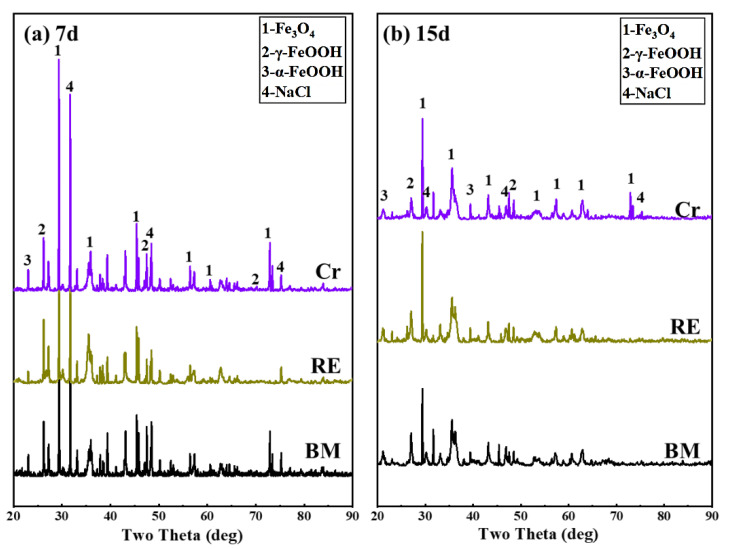
Phase compositions of the corrosion products formed on the three steel rebars after corrosion tests characterized by XRD: (**a**) 7 days and (**b**) 15 days.

**Table 1 materials-15-08269-t001:** Fitted results of EIS data for the passive film on steel rebars.

Steels	Solution	*R_ct_*, kΩ·cm^2^	*Q* _dl_		*R*_*f*_, kΩ·cm^2^	*Q* _f_	
			Y_0_ Ω^−1^·cm^2^·s^n^	n		Y_0_ Ω^−1^·cm^2^·s^n^	n
BM	0 Cl	61.50	1.26 × 10^−5^	0.78	72.39	1.82 × 10^−5^	0.84
	0.1 Cl	26.08	8.45 × 10^−6^	0.61	33.17	2.76 × 10^−5^	0.94
	1Cl	20.31	4.91 × 10^−5^	0.92	23.56	1.22 × 10^−4^	0.74
	3.5 Cl	14.59	5.13 × 10^−5^	0.93	18.23	9.38 × 10^−5^	0.78
Cr	0 Cl	77.80	5.30 × 10^−6^	0.62	88.31	2.89 × 10^−5^	0.93
	0.1 Cl	35.21	1.64 × 10^−5^	0.88	43.74	2.03 × 10^−5^	0.89
	1 Cl	28.56	5.95 × 10^−5^	0.91	29.67	6.09 × 10^−5^	0.87
	3.5 Cl	11.73	4.56 × 10^−5^	0.88	21.53	1.11 × 10^−5^	0.98
RE	0 Cl	46.73	3.69 × 10^−8^	1.00	55.78	4.92 × 10^−5^	0.91
	0.1 Cl	25.33	5.18 × 10^−5^	0.93	32.46	8.64 × 10^−6^	0.75
	1 Cl	20.72	7.55 × 10^−5^	0.91	26.19	2.32 × 10^−4^	0.79
	3.5 Cl	6.07	5.74 × 10^−8^	0.92	10.50	1.20 × 10^−4^	0.89

## Data Availability

Not applicable.

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
