# Peer review of "Attempt to Optimize the Corrosion Resistance of HRB400 Steel Rebar with Cr and RE"

_materials, 2022, doi:10.3390/ma15228269_

Round 1

Reviewer 1 Report

The article is devoted to the study of the corrosion resistance of the HRB400 steel rebar alloyed with Cr and rare earths (RE). The authors describe the effect of Cr and RE on the microstructure, the morphology of inclusions, localized corrosion morphology, surface and cross-sectional morphology of the rust layer, the elemental valence of the passivation film, the phase of the rust and the electrochemical properties.

Some remarks:

1. The abbreviation RE is not elucidated in the abstract.

2. Line 114: the abbreviation BM is not entered.

3. Were BM, Cr and RE steels smelted by the same method? What? What is the mode of their heat treatment before corrosion testing?

4. Fig. 2c shows a rather large inclusion. What is the volume fraction of such inclusions? Do they have an internal structure or is it a conglomerate of particles?

5. Line 153: “…The corrosion pits on the surface of the RE bar are the smallest…”

What are they smaller compared to? On fig. 3c pits look larger than in Fig. 3 a and b.

6. Have statistical data been obtained on the size of corrosion pits for different steels?

7. Fig. 4c does not show a map for O.

8. There are different rulers in the SEM images. It makes it difficult to compare the sizes of structural elements.

9. Line 302: Fig. 2 should probably be replaced by Fig. 11.

10. Line 341: Fig. 2 should probably be replaced by Fig. 12.

11. Line 370: Fig. 3 should probably be replaced with Fig. 13. Please check the numbering of figures in the text.

12. It is interesting to compare how oxide layers form on the surface of steel in the areas of ferrite and pearlite. Was any difference found?

Author Response

The article is devoted to the study of the corrosion resistance of the HRB400 steel rebar alloyed with Cr and rare earths (RE). The authors describe the effect of Cr and RE on the microstructure, the morphology of inclusions, localized corrosion morphology, surface and cross-sectional morphology of the rust layer, the elemental valence of the passivation film, the phase of the rust and the electrochemical properties.

Some remarks:

Thank you for the professional comments. Based on the comments, all revised contents have been marked in red in the manuscript.

  1. The abbreviation RE is not elucidated in the abstract.

Thanks for the critical suggestion, and now the abbreviation RE has been revised as the rare earths (RE) in the abstract.

  1. Line 114: the abbreviation BM is not entered.

Based on the comment, the base material (BM) has been added in the manuscript.

  1. Were BM, Cr and RE steels smelted by the same method? What? What is the mode of their heat treatment before corrosion testing?

Yes, all steels were smelted by the same method finished by the Wuhu-Xinxing pipes Co. LTD. There was no special pre-heat treatment for steels before the corrosion tests.

  1. Fig. 2c shows a rather large inclusion. What is the volume fraction of such inclusions? Do they have an internal structure or is it a conglomerate of particles?

Thank you for this important comment. In this work, the element distribution and the shape were considered, without the value fraction of this inclusion. Based on previous work [1-3], the internal structure can be obtained by the electrolysis method, which presents an irregular spheroid.

[1] R. Avci, B.H. Davis, M.L. Wolfenden, I.B. Beech, K. Lucas, D. Paul. Mechanism of MnS-mediated pit initiation and propagation in carbon steel in an anaerobic sulfidogenic media. CORROS SCI. 76 (2013), 267-274.

[2] C. Liu, X. Cheng, Z. Dai, R. Liu, Z. Li, L. Cui, M. Chen, L. Ke. Title Synergistic Effect of Al2O3 Inclusion and Pearlite on the Localized Corrosion Evolution Process of Carbon Steel in Marine Environment. MATERIALS. 11 (2018), 2277.

[3] C. Liu, R.I. Revilla, Z. Liu, D. Zhang, X. Li, H. Terryn. Effect of inclusions modified by rare earth elements (Ce, La) on localized marine corrosion in Q460NH weathering steel. CORROS SCI. 129 (2017), 82-90.

  1. Line 153: “…The corrosion pits on the surface of the RE bar are the smallest…”

What are they smaller compared to? On fig. 3c pits look larger than in Fig. 3 a and b.

Thank you for the comment. It is a mistake that the value surveyor's rod in Fig. 3c was set as 10 μm. In fact, the value should be 600 nm, which was consistent the results in Fig. 4c.

  1. Have statistical data been obtained on the size of corrosion pits for different steels?

The statistical data of corrosion pits were absent due to the similar corrosion results in the macro-morphology. The function of RE and Cr should be clarified through the observation under micro scale.

  1. Fig. 4c does not show a map for O.

Yes, the O element was not detected in the corrosion pits. The complex inclusion in the pits is composed of the (RE)2O2S and the MnS, and absence of O should be related to the dissolution of (RE)2O2S [1, 2].

[1] C. Liu, R.I. Revilla, Z. Liu, D. Zhang, X. Li, H. Terryn. Effect of inclusions modified by rare earth elements (Ce, La) on localized marine corrosion in Q460NH weathering steel. CORROS SCI. 129 (2017), 82-90.

[2] C. Liu, H. Yuan, X. Li, Z. Che, S. Yang, C. Du. Initiation Mechanism of Localized Corrosion Induced by Al2O3-MnS Composite Inclusion in Low-Alloy Structural Steel. METALS-BASEL. 12 (2022), 587.

  1. There are different rulers in the SEM images. It makes it difficult to compare the sizes of structural elements.

Thank you for the professional suggestion. It is a mistake to make the unactual ruler in the SEM images. Through comparing with the original graphs, all rulers have been revised in the results.

  1. Line 302: Fig. 2 should probably be replaced by Fig. 11.
  2. Line 341: Fig. 2 should probably be replaced by Fig. 12.
  3. Line 370: Fig. 3 should probably be replaced with Fig. 13. Please check the numbering of figures in the text.

Thank you for the 9-11 comment. The number of figures has been revised in the corresponding position in the manuscript.

  1. It is interesting to compare how oxide layers form on the surface of steel in the areas of ferrite and pearlite. Was any difference found?

Yes. Due to the existence of pearlite in matrix, the galvanic corrosion effect between ferrite and pearlite, and the oxide layer on ferrite/pearlite are different. The inhomogeneous oxide layer on the whole steel surface presents different properties, which can be evaluated based on the EIS results. The addition of Cr increased the proportion of ferrite, leading to the more protective oxide layer on steel surface than BM.

Reviewer 2 Report

the topic is interesting and in most of the cases tests are well describe. In any case results are not consistent with the well known theory of corrosion in concrete. I strongly suggest to improve the paper. 

Author Response

Comments and request to revision:

References:

  • too much self-citation and un-appropriate cited paper. For example, why to cite [1] which is not pertinent with carbon steel in concrete??

Thank you for the professional comment. Some unsuitable references are replaced with the relevant articles, which are marked in red in References.

  • Please refer to standard for the tests

The tests were conducted on sample based on previous studies, and the numbers of related references have been added in the manuscript.

  • No citations related to chloride-induced corrosion in concret, at least on the critical chloride threshold

The citations of related contents have been added in the References:

[1] R. Liu, L.H. Jiang, J. Xu, C. S. Xiong, Z. J. Song. Influence of carbonation on chloride-induced reinforcement corrosion in simulated concrete pore solutions. CONSTR BUILD MATER. 56 (2014), 16-20

[19] M. Liu, X.Q. Cheng, X.G. Li, Z. Jin, H. X. Liu. Corrosion behavior of Cr modified HRB400 steel rebar in simulated concrete pore solution. CONSTR BUILD MATER. 93 (2015), 884-890

Experimental

  • More detailed on the electrochemical tests are mandatory: applied standard, equilibrium time, potential scan rate, frequency in the case of EIS, potential amplitude.

Based on the suggestion, the contents about the electrochemical tests have been added in the manuscript, which can be seen as follows:

Potentiodynamic polarization tests were conducted from −1 to 1 VSCE at a scanning rate of 0.1667 mV/s. A stabilization time of 50 min was allowed prior to EIS measurement. EIS measurements were performed over the frequency range from 100 kHz to 10 mHz with a 10 mV sinusoidal amplitude at open-circuit potential (OCP), and the data were fitted with ZSimpWin software [20, 21].

  • It is well known that test performed by adding chloride to the alkaline solution during carbon steel passivation are not correct to study corrosion in concrete. Chloride in fact interfere with the passive film formation. So, during passive formation the effect of Cr or RE can be indered or enhanced by the chlorides. Chloride must be added after passivation. Why the authors decided to add chlorides at the very beginning? and above all, are the results reliable? and applicable to real cases?

Yes. Adding Cl- simulated concrete interstitial fluid in alkaline solution does have some problems, but it is undeniable that this method is widely used at present, which can quickly evaluate the corrosion resistance of the reinforcement body to a certain extent, especially in the reinforced concrete structure made of sea sand. [1-3]. As the author said, Cl- is indeed an important factor affecting the passive film. We added Cl- at the beginning to know the difference of corrosion resistance of materials in some harsh environments when the passive film on the surface of reinforcement has not been formed, which is also a common method in literature review [4-6].

In this work, the protectiveness of passive film against chloride ion was evaluated by the Cyclic potentiodynamic polarization tests, and the passive current density presented different values in the whole tests. The solution without Cl- was selected as the blank experiment, as shown in Fig. 6a. With the increase of Cl- content, we found that the passivation process was blocked and the passivation range was shortened.

[1] M. Liu, X. Cheng, X. Li, Z. Jin, H. Liu. Corrosion behavior of Cr modified HRB400 steel rebar in simulated concrete pore solution. CONSTR BUILD MATER. 93 (2015), 884-890.

[2] Y. Tian, M. Liu, X. Cheng, C. Dong, G. Wang, X. Li. Cr-modified low alloy steel reinforcement embedded in mortar for two years: Corrosion result of marine field test. Cement and Concrete Composites. 97 (2019), 190-201.

[3] M. Liu, X. Cheng, X. Li, Y. Pan, J. Li. Effect of Cr on the passive film formation mechanism of steel rebar in saturated calcium hydroxide solution. APPL SURF SCI. 389 (2016), 1182-1191.

[4] M. Liu, X. Cheng, X. Li, C. Zhou, H. Tan. Effect of carbonation on the electrochemical behavior of corrosion resistance low alloy steel rebars in cement extract solution. CONSTR BUILD MATER. 130 (2017), 193-201.

[5] M. Liu, X. Cheng, X. Li, T.J. Lu. Corrosion behavior of low-Cr steel rebars in alkaline solutions with different pH in the presence of chlorides. J ELECTROANAL CHEM. 803 (2017), 40-50.

[6] M. Liu, X. Cheng, X. Li, J. Hu, Y. Pan, Z. Jin. Indoor accelerated corrosion test and marine field test of corrosion-resistant low-alloy steel rebars. CASE STUD CONSTR MAT. 5 (2016), 87-99.

  • Immersion test last few minutes or hours. The passivation process takes 2 days to reach a steady stable condition. Why the authors selected a so short test duration? can the results be considered reliable at the same time?

This is a professional comment. Due to the small addition of RE, the immersion tests were conducted on samples without considering the stabilization of passive film, and focused on the functions of RE on the pitting corrosion behavior. The effects of Cr on the properties of passive film were characterized by electrochemical and XPS tests.

  • Pit measurement. The most important information is missing. No indication is given of pit depth. Please add this point since it is very fundamental in verifying the quality of a passive layer

Yes. The pit measurement is critical for comparing the quality of passive film. While, in this work, we used the XPS technique for surveying the compounds of Fe, Cr and O in passive film, which is more fontal realness to demonstrating the quality of passive film on steel, which can be seen in Fig. 7.

  • As a consequence of the previous bullet, how much is it the local corrosion rate? how it was estimated?

The comment is professional. The corrosion rate was hardly to be estimated for the pitting corrosion. In this work, the effects of RE and Cr on the corrosion resistance of steel rebars were discussed considering comprehensive impacts of the formation of corrosion pits, the stabilization of passive film and the protectiveness of rust layer, which can fully reveal the functions of RE and Cr on the corrosion resistance of steel rebars.

Results

  • Potentiodynamic tests. The value of the current in the passive interval cannot be defined as passive current density but as anodic current density in the passive range. In other words, is it not correct to use potentiodynamic tests to define (and discuss) the passive current density? The patter is typically measured with potentiostatic tests, or better with immersion tests in freely corroding condition.

Yes. The passive current density during the passive range reflects a dynamic balance process, including the formation of passive film and the dissolution of passive film. In this work, both potentiodynamic tests and EIS measurements were conducted on samples for discussing the protectiveness of passive film.

Based on the constructive comment from the reviewer, the current density under 0.4V voltage is calculated uniformly in Fig. 6a and b (unsignificant differences among steels), and the obtained results have been added in the manuscript.

  • Figure 6: indicate the units for chloride content

The unit wt. % has been added in the Figure 6 caption.

  • Figure 11. Corrosion rates in the range 0.3 to 3 mm/a are very very high, it is well known that in passive condition the corrosion rate MUST be lower than 0,002 mm/a. These data seem to invalidate all the discussion. How you can consider acceptable a valus as high as 0.3 mm/a

Thank you for this critical comment. In this work, we used the accelerating dry/wet cyclic tests, whose experimental parameters were harder than the alkalescency condition without Cl- for steel rebars, resulting in the more serious corrosion. Thus, the corrosion rates present higher values than the 0.002 mm/a. While, the accelerating corrosion tests can effectively discuss the functions of Cr and RE on the protectiveness of rust layer. So this is an accelerated test, which will corrode at a much faster rate than the actual service environment [1-3].

[1] Z. Liu, W. Hao, W. Wu, H. Luo, X. Li. Fundamental investigation of stress corrosion cracking of E690 steel in simulated marine thin electrolyte layer. Corros. Sci., 148 (2019), pp. 388-396

[2] T. Zhu, F. Huang, J. Liu, Q. Hu, W. Li. Effects of inclusion on corrosion resistance of weathering steel in simulated industrial atmosphere. Anti Corros. Methods Mater., 63 (2016), pp. 490-498

[3] W. Wu, Z. Dai, Z. Liu, C. Liu, X. Li. Synergy of Cu and Sb to enhance the resistance of 3%Ni weathering steel to marine atmospheric corrosion. Corros. Sci., 183 (2021), Article 109353

Reviewer 3 Report

In this work, the effect of Cr and RE on the protectiveness of the passive film and the rust layer 392 on the HRB400 steel rebar (BM) has been investigated. 

The work is well-designed and presented. The work is recommended for publication after careful revision/review by the authors.

Author Response

Thank the reviewer for this approval.